

# [230]Th/U Isochron Dating of Cryogenic Cave Carbonates

Paul Töchterle[1], Simon D. Steidle[1], R. Lawrence Edwards[2], Yuri Dublyansky[1], Christoph Spötl[1], Xianglei Li[2], John Gunn[3], Gina E. Moseley[1]

[1]Institute of Geology, University of Innsbruck, Innsbruck, 6020, Austria
[2]Dept. Earth and Environmental Sciences, University of Minnesota, Minneapolis, 55455, MN/USA
[3]School of Geography, Earth and Environmental Sciences, University of Birmingham, Birmingham, B15 2TT, UK

*Correspondence to*: Paul Töchterle (paul.toechterle@uibk.ac.at)

**Abstract.** Cryogenic Cave Carbonates (CCCs) are a type of speleothem, typically dated with [230]Th/U disequilibrium methods, that provide evidence of palaeo-permafrost conditions. In the field, CCCs occur as distinct patches of millimetre- to centimetre-
sized loose crystals and crystal aggregates on the floor of cave chambers, lacking a framework to validate ages by stratigraphic order. Correction factors for initial [230]Th ($^{230}Th_0$) are often based on the bulk-earth derived ratio of initial [230]Th/[232]Th activity (($^{230}Th/^{232}Th)_0$), which is a well-established approach when $^{230}Th_0$ is moderately low. For samples with elevated levels of $^{230}Th_0$, however, accuracy can be improved by constraining $(^{230}Th/^{232}Th)_0$ independently. Here, we combine detailed morphological observations from three CCC patches found in Water Icicle Close Cavern in the Peak District (UK) with [230]Th/U
analyses. We find that individual CCC crystals show a range of morphologies that arise from non-crystallographic branching in response to the chemical evolution of the freezing solution. Results of [230]Th/U dating indicate that samples with a large surface area relative to the sample volume are systematically more affected by contamination with $^{230}Th_0$. By fitting isochrons to these results, we test whether the CCCs in an individual patch formed during the same freezing event, and demonstrate that $(^{230}Th/^{232}Th)_0$ can deviate substantially from the bulk-earth derived value and also vary between the different CCC patches.
Where CCCs display elevated $^{230}Th_0$, isochrons are a useful tool to constrain $(^{230}Th/^{232}Th)_0$ and obtain ages with improved accuracy. Detritus absorbed to the crystal surface is shown to be the most likely source of $^{230}Th_0$. Our results suggest that some previously published CCC ages may merit re-assessment and we provide suggestions on how to approach future dating efforts.

## 1 Introduction

Cryogenic Cave Carbonates (CCCs) are speleothems that typically occur as an accumulation of loose crystals on the floor of
cave chambers. Despite being described in earlier literature (Skřivánek, 1954), it was not until the early 2000s that their value as palaeoclimate archives became apparent (Žák et al., 2004). The widely accepted model of CCC formation involves precipitation in slowly freezing pools of water on top of cave ice bodies. This particular setting requires pre-existing, sizeable bodies of cave ice with cave air temperatures between approximately -1 and 0 °C (i.e. the 'CCC window', Spötl et al., 2021; Koltai et al., 2021), indicating that the cave was surrounded by permafrost at the time of CCC formation. A study of Winter





Wonderland Cave, USA (Munroe et al., 2021), validated this genetic model by observing very recent formation of CCCs in the exact setting as proposed by Žák et al. (2004).

As with common speleothems (e.g. stalagmites and flowstones), the formation age of CCCs is determined by $^{230}$Th/U disequilibrium dating. The principle of this method relies on the large solubility difference between uranium and thorium in water (Gascoyne, 1992). Age precisions as low as 2‰ are routinely achieved using the latest analytical protocols (Cheng et

al., 2013). However, the accuracy of dating results can be compromised if substantial amounts of thorium incorporated at the time of formation ($^{230}$Th$_0$) are not adequately corrected for (Dorale et al., 2004). In practice, $^{230}$Th$_0$ is identified and corrected by measuring the chemically equivalent $^{232}$Th (expressed herein as measured ratio ($^{230}$Th/$^{232}$Th)$_t$; parenthesis signify activity ratio hereafter), a long-lived isotope that is not part of the $^{238}$U decay chain. Subsequently, $^{230}$Th$_0$ is subtracted by applying an initial $^{230}$Th/$^{232}$Th activity ratio (($^{230}$Th/$^{232}$Th)$_0$). However, in nature, values for ($^{230}$Th/$^{232}$Th) vary by orders of magnitude (from

<1 to >50, e.g., R. Scott, 1968; Moore and Sackett, 1964; Moore, 1981; Hubert et al., 2006; Hirose et al., 2012). Consequently, ($^{230}$Th/$^{232}$Th)$_0$ needs to be accurately constrained if a sample's $^{230}$Th$_0$ is high enough to make the age equation sensitive to the correction term. There are four widely used approaches to derive ($^{230}$Th/$^{232}$Th)$_0$: (1) by using the bulk-earth composition (e.g. Wedepohl, 1995); (2) by deriving it from stratigraphic constraints (e.g. Cheng et al., 2000; Hellstrom, 2006); (3) by using modern values measured at the field site, for example in groundwater; (4) by calculating values derived from isochrons (e.g.,

Lin et al., 1998; Henderson and Slowey, 2000; Cobb et al., 2003).

Since the early 2000's, more than 20 studies published $^{230}$Th/U ages of CCCs, commonly from multiple samples, and interpreted them with regard to regional permafrost conditions (e.g., Richter and Riechelmann, 2008; Žák et al., 2009; Richter et al., 2010; Žák et al., 2012; Orvošová et al., 2014; Spötl and Cheng, 2014; Chaykovskiy et al., 2014; Richter et al., 2018; Koltai et al., 2021; Spötl et al., 2021). However, reported ages from a given patch of CCCs were rarely synchronous and

sometimes spanned thousands of years and different climatic events (Luetscher et al., 2013; Dublyansky et al., 2018). This spread of ages led investigators to conclude that in those cases, CCCs must have formed during multiple freezing events. Since CCCs typically occur as accumulations of crystals with no internal stratigraphy, it is generally not possible to constrain ($^{230}$Th/$^{232}$Th)$_0$ by stratigraphic order, validate individual ages or detect potential outliers and/or erroneous measurements (i.e., 'age reversals'), hence, in almost all cases, $^{230}$Th$_0$ was corrected by assuming a bulk-earth composition for ($^{230}$Th/$^{232}$Th)$_0$.

In this study, we test the hypothesis that CCCs from a single patch formed coevally within achievable dating precision. We present a comprehensive dataset of three CCC patches from Water Icicle Close Cavern comprising 26 $^{230}$Th/U ages, carbonate stable isotope data, and detailed morphological observations. For the first time, we apply an isochron approach to constrain ($^{230}$Th/$^{232}$Th)$_0$ in CCCs in order to test whether an observed spread in ages is the result of multiple generations of CCC being present in a patch or arises from variable amounts of $^{230}$Th$_0$ across different samples.





## 2 Study Site

The samples used in this study were collected from Water Icicle Close Cavern (WICC), a limestone dissolution cave located in the Peak District (United Kingdom). The cave entrance (53.1781 °N, 1.7605 °W) is situated on a plateau at 338 m above sea level and is surrounded by a low-gradient hilly topography that has been incised by valleys, most of which are dry. The bedrock geology comprises a Lower Carboniferous limestone sequence, several hundred metres thick. In total, the cave comprises over 1000 m of explored passages that formed along a horizontal level 30-40 m below the surface. The climate of the region is temperate with a mean annual air temperature of 8.9 °C and 1160 mm of annual precipitation (UK METoffice, 2020).

Previous research reported evidence of extensive former ice presence in the cave including speleothem damage, ice attachments and solifluction deposits (Gunn et al., 2020). Dating of several broken and re-sealed speleothems points towards at least one period of cave glaciation between 87,000 and 83,000 years ago that can be related to the presence of permafrost. The area around Water Icicle Close Cavern was not glaciated during the Last Glacial Maximum but the British-Irish Ice Sheet terminated only around 30 km northwest of the cave at its maximum extent (Clark et al., 2012). Overall, 11 patches of CCC were documented in the cave, three of which were sampled in the course of this study.

## 3 Methods

### 3.1 Sampling Strategy

Three patches of CCCs were sampled according to different strategies depending on the respective grain size of the occurrence. If the mean crystal size was larger than approximately 5 mm, several individual crystals were selected while trying to obtain a representative subsample of all morphological varieties of the patch. For smaller grain sizes, a subsample was simply scooped up with a knife. With all sampling, damage to the cave was minimised such that the overall appearance of the CCC patches was kept intact.

### 3.2 Carbonate Stable Isotope Analysis

While CCCs can be identified by their characteristic crystal morphology and field occurrence, the diagnostic property separating them from common speleothems is the stable isotope composition (Žák et al., 2018). Several morphological varieties of each sampled CCC patch were selected for stable isotope analysis. If grain size permitted, aliquots were taken from the same specimen that was also used for petrographic analysis.

Samples were cleaned in an ultrasonic bath of de-ionised water before drilling between 0.10 and 0.80 mg of carbonate powder from random points on the surface with a carbide burr-tipped dental drill. The stable isotope composition of these carbonate powders was then determined using a Thermo Scientific Delta V Plus isotope ratio mass spectrometer coupled to a GasBench





II (Spötl and Vennemann, 2003). An analytical uncertainty (1σ) of ±0.06‰ for $\delta^{13}C$ and ±0.08‰ for $\delta^{18}O$ applies to all stable isotope data (Spötl, 2011) and values are reported relative to the Vienna PeeDee Belemnite (VPDB) standard.

### 3.3 $^{230}$Th/U analysis

As with stable isotope analyses, smaller crystals were rinsed with deionised water for several minutes in an ultrasonic cleaner. Visible surficial detritus was removed with a toothbrush. Samples were then processed as a whole. For two large samples of patch 3 (PT25 and PT26, see table 1) it was possible to cut them in half using a diamond-coated wire saw, polish and drill an aliquot form the inner part of the sample of approximately 30 mg with a burr-tipped dental drill in a laminar-flow hood.

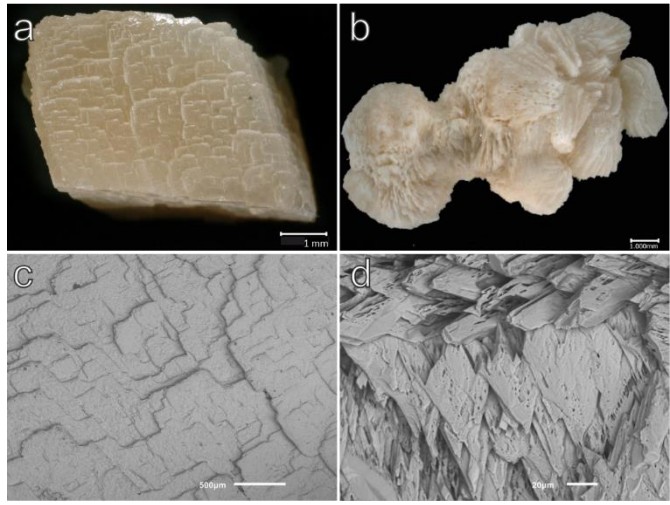

**Figure 1:** Selected CCC morphologies of patch 1. a) Rhombohedral calcite crystal with stepped surface, b) complex intergrown crystals composed of radial-fibrous calcite, c) SEM image of the surface of the sample depicted in a), d) SEM image of b).

Chemical preparation and mass spectrometry were performed at the Trace Metal Isotope Geochemistry Laboratory at the University of Minnesota. Elemental extraction of U and Th was performed according to the procedures described in Edwards et al. (1987). The resulting solutions were measured on a Thermo Fisher Neptune Plus multi-collector inductively coupled plasma mass spectrometer in peak-jumping mode on a secondary electron multiplier (Shen et al., 2012).

Ages were calculated according to Edwards et al. (1987) with decay constants for uranium $\lambda_{238}$ = 1.55125 *$10^{-6}$ (Jaffey et al., 1971), $\lambda_{234}$ = 2.82206 *$10^{-6}$ and thorium $\lambda_{230}$ = 9.1705 *$10^{-6}$ (Cheng et al., 2013). Measurement uncertainties are given at the 2σ level and ages are reported as years before 1950 CE (BCE).

Each uncorrected age was subsequently corrected for $^{230}$Th$_0$ based on a range of ($^{230}$Th/$^{232}$Th)$_0$ values, starting from a bulk-earth value of 0.82 ± 0.41, representing secular equilibrium with a ($^{232}$Th/$^{238}$U) value of 3.8 (Wedepohl, 1995), to 15 times this bulk-earth ratio (i.e.: ($^{230}$Th/$^{232}$Th)$_0$ = [0.82 ± 0.41, ... , 12.23 ± 0.41], precision arbitrarily assumed as 50% of bulk-earth). The resulting array of corrected ages and their respective uncertainties was converted to probability density functions (PDFs) assuming a Gaussian distribution and summed per patch in order to compare with isochron dating results.

Isochrons were constructed by applying least squares regressions in a three-dimensional space of ($^{234}$U/$^{238}$U), ($^{232}$Th/$^{238}$U) and ($^{230}$Th/$^{238}$U) (Ludwig and Titterington, 1994). Isochron ages are calculated from the ($^{234}$U/$^{238}$U) and ($^{230}$Th/$^{238}$U) intercept ± the 95% confidence interval of the respective regression.



### 3.4 Morphological Analysis

The cleaned samples were examined using a Keyence
VHX-6000 digital microscope. To investigate the
surface texture of CCCs, selected samples were gold-
coated and analysed using a JEOL JSM-6010LV
scanning electron microscope. Secondary electron
images were acquired using 15 kV accelerating
voltage.

## 4 Results and Discussion

### 4.1 Sample Characterisation

The sampled CCC patches in Water Icicle Close
Cavern are composed of calcite. Grain size varies
within each patch, but most specimens are
approximately 1-10 mm in diameter. One out of the
three sampled patches also contains larger crystals up

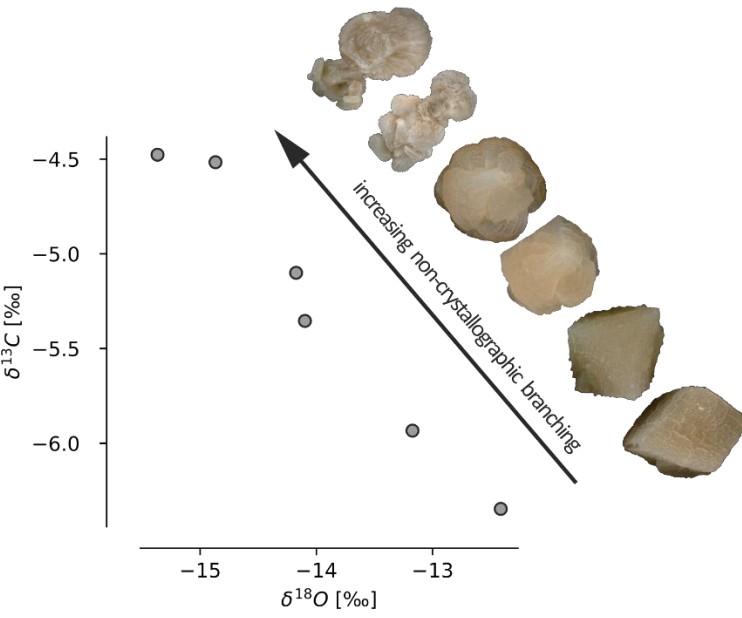

**Figure 2:** Stable isotope composition of selected crystals from patch 1. The data follow a trend with a negative slope, that is typical for CCC. The stable isotope composition changes systematically with the progression from rhombohedral to radially branching morphologies. All depicted samples are between 2 and 5 mm in diameter, but images were rescaled for better comparability.

to 3 cm in diameter. Each occurrence comprises a variety of crystal morphologies, as is typical for CCCs (Žák et al., 2018).
Prevalent shapes include rhombohedra (fig. 1a) and aggregates of crystals (fig. 1b). Some specimens comprise elongated sheets
or needles branching radially from a common centre point and show an almost spherulitic shape. Secondary electron
microscopy (SEM) images reveal that rhombohedra-like morphologies have comparably smooth, stepped surfaces that mimic
the overall rhombohedral geometry (fig. 1c). More complex grains on the other hand have a highly irregular surface but
elongated rhombohedral shapes can still be recognised (fig. 1d).

The stable isotope composition of CCCs from patch 1 varies systematically with crystal morphology (fig. 2). Rhombohedral
crystals have $\delta^{18}O$ values from -12.4 to -14.1 ‰ and $\delta^{13}C$ values from -6.4 to -5.4 ‰. In comparison, complex grains show
lower $\delta^{18}O$ (-14.2 to -15.4 ‰) and higher $\delta^{13}C$ values (-5.1 to -4.5 ‰).

These observations can be interpreted in the context of the genetic model of CCCs (Žák et al., 2004). As water freezes, the
$\delta^{18}O$ of residual water lowers progressively because $^{18}O$ is preferentially incorporated into the growing ice. Conversely, $\delta^{13}C$
increases due to preferential loss of $^{12}C$ as $CO_2$ degasses. This trend is well established for CCCs (Žák et al., 2008; Žák et al.,
2018) and is also observed from core to rim within single CCC grains (Töchterle, 2018).

The concept of a morphological evolution of CCCs has been formulated in previous studies (Žák et al., 2012), but analytical
evidence supporting it has been lacking. The observed correlation between crystal morphology and stable isotope composition





(fig. 2) indicates that these CCCs formed during a single freezing event. Furthermore, complex aggregates apparently formed later in the freezing
process than rhombohedral grains.

During freezing, it is not only the stable isotope composition that changes, but also the overall concentration of solutes. A large body of literature confirms the effects of thermodynamic variables such as supersaturation, ionic strength and the concentration of foreign ions or organic compounds
on calcite morphologies and fabrics (e.g. Frisia et al., 2000; Sunagawa, 2005; Sand et al., 2012; Frisia, 2015; Hong et al., 2016; Mercedes-Martín et al., 2021). The morphological variety of CCCs observed in this study can be viewed as an expression of changing crystallisation driving forces in response to progressive freezing. Two fundamental pathways can alter the
shape of a grain. Firstly, multiple individual crystals can intergrow to form crystal aggregates. Secondly, the crystal fabric can be influenced by various degrees of non-crystallographic branching in response to fast growth and/or high supersaturation of the parent solution, resulting in radial growth with the endmember being a spherulite (Sunagawa, 2005; Shtukenberg et al.,
2012; Yoreo et al., 2015). Crucial for this study, by building a more complex geometry or creating a rougher surface, both of these mechanisms potentially increase the relative surface area of a grain (i.e., the ratio between surface area and volume).

**Figure 3:** Uncorrected ages plotted against the respective $(^{230}Th/^{232}Th)_t$ ratio. Colour represents morphology type. Also shown for each patch are corresponding isochron ages (±95 % confidence interval) and the summed normalised probability density functions (PDFs), calculated from an array of $(^{230}Th/^{232}Th)_0$ values. Samples of type A in patch 2, which are much older, are not shown for better scaling.

### 4.2 $^{230}$Th/U Analyses

Overall, 26 $^{230}$Th/U analyses were carried out on CCCs from three patches (table 1). From each patch, three samples of three morphological types were selected. All analyses yielded high concentrations of $^{238}$U (1035 ±1 to 3843 ±5 ng g$^{-1}$) and mostly low concentrations of detrital thorium (expressed as higher values of $(^{230}Th/^{232}Th)_t$). In the following sections the results from
each patch will be discussed individually. Samples are grouped and sorted into morphological types (A, B and C) according to the relative surface area and roughness on a visual basis.

### 4.2.1 Patch 1

The samples selected from patch 1 comprise rhombohedral single crystals (morphology A), branched aggregates (morphology B), and also a set of transitional forms that show a progression from lower to higher degrees of non-crystallographic branching





(morphology C, suppl. fig. 1). In this patch, $(^{230}Th/^{232}Th)_t$ varies by more
than one order of magnitude (from 7.7 ±0.2 to 150 ±3) and correlates with
the morphology type whereby the complex morphologies B and C show
lower $(^{230}Th/^{232}Th)_t$ values (table 1) indicative of higher $^{230}Th_0$. More
complex morphologies therefore yielded older uncorrected ages. The

difference between these uncorrected ages is as large ~6,200 years (fig 3).
Given the substantial spread in nominal ages and lack of constraints on
$(^{230}Th/^{232}Th)_0$, it is tempting to infer distinct formation events for the
different morphological types. However, CCCs in this patch conform to a
morphological evolution as described in section 4.1, implying that they

likely formed from the same parent solution during a single freezing event.
It seems unlikely, however, that such an event lasted several thousand years,
and that the cave temperature remained stable throughout this entire period.
The question arises if it is possible to find a $(^{230}Th/^{232}Th)_0$ ratio that would
result in overlapping corrected ages, supporting the hypothesis that all

samples formed at the same time and thus reconcile dating results with
morphological observations.

We calculated an array of corrected ages based on multiple possible
$(^{230}Th/^{232}Th)_0$ ratios (30 ratios between 1x and 15x bulk-earth) and derived
a summed PDF (shaded area in fig 3), representing a cumulative likelihood

distribution of all corrected ages in this patch. Naturally, samples with low
$(^{230}Th/^{232}Th)_t$ are highly sensitive to $(^{230}Th/^{232}Th)_0$ and yielded very wide
distributions compared to samples with $(^{230}Th/^{232}Th)_t > 100$.

Figure 4 shows the $^{238}U$-normalised activities of $^{230}Th$ and $^{232}Th$. These
variables correlate with an R² of 0.98 (r1.1 in fig. 4, regression parameters

in suppl. table 1), meaning that 98 % of the observed variance in
$(^{230}Th/^{238}U)$ (which mainly determines the ages of these samples) can be
explained by different levels of $^{232}Th$. From the least-squares regression, it

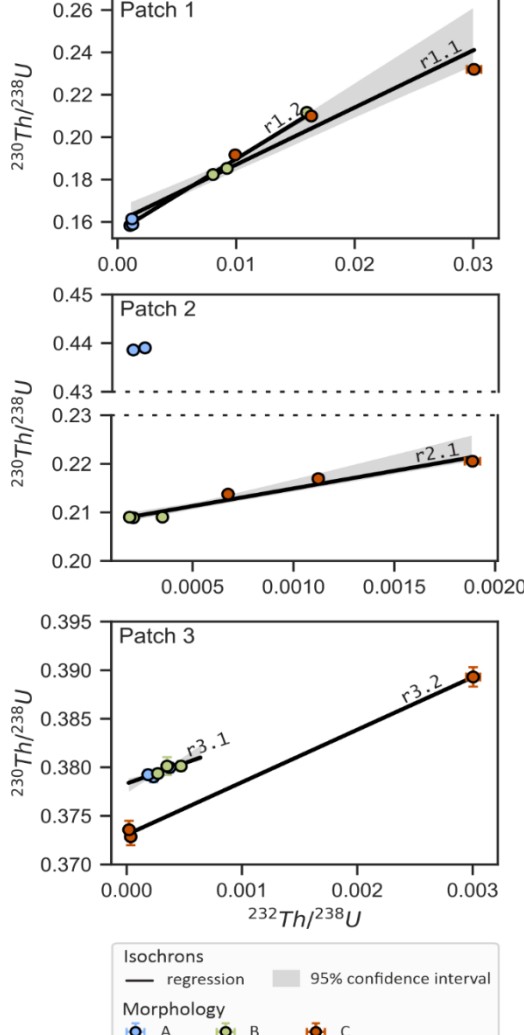

**Figure 4:** Isochron plots of $^{238}U$-normalised thorium measurements of each CCC patch. Colour represents morphological type. The shaded area represents the 95% confidence interval of the regression slope and intercept. Regression lines are labelled with an identification number corresponding to supplementary table 1.

is possible to calculate an isochron age (12,224 ± 478 BCE) and a $(^{230}Th/^{232}Th)_0$ activity ratio of 2.7 ±0.2. This result partly
overlaps the peak of the PDF. However, the linear regression model strongly depends on the oldest uncorrected age (PT33).

When disregarding PT33, the isochron age shifts to 11,865 ± 172 BCE (r1.2, R² = 0.997) with a $(^{230}Th/^{232}Th)_0$ of 3.4 ±0.1,
improving the agreement between the isochron and PDF ages. The isochron age of r1.2 also intercepts high-$(^{230}Th/^{232}Th)_t$
samples notably better, which are less sensitive to differences in the $(^{230}Th/^{232}Th)_0$ values. Excluding samples with very high





levels of $^{230}Th_0$ may be justified in cases where multiple sources of $^{230}Th_0$ were mixed with varying ratios, as will be discussed in section 4.3.

**Table 1:** Results of $^{230}Th/U$ analyses of CCC from Water Icicle Close Cavern. Uncertainties are reported at the 2σ level.

| Patch | Morph. Type | ID | $^{238}U$ [ng g$^{-1}$] | $^{232}Th$ [pg g$^{-1}$] | $(^{230}Th/^{232}Th)_t$ | $\delta^{234}U^*$ | $(^{230}Th/^{238}U)$ | $^{230}Th$ age [BCE] uncorrected | $^{230}Th$ age [BCE] isochron corrected | $\delta^{234}U_{initial}^*$ | isochron $(^{230}Th/^{232}Th)_0$ |
|---|---|---|---|---|---|---|---|---|---|---|---|
| 1 | A | PT27 | 1466 ±1 | 4733 ±95 | 150 ±3 | 489.6 ±2.0 | 0.1583 ±0.0003 | 12110 ±29 | 11850 ±30 | 506.3 ±2.0 | |
| | A | PT28 | 1487 ±1 | 5510 ±110 | 133 ±3 | 488.7 ±1.5 | 0.1614 ±0.0002 | 12369 ±24 | 12071 ±27 | 505.8 ±1.5 | |
| | A | PT29 | 1680 ±3 | 6514 ±131 | 125 ±3 | 492.8 ±2.5 | 0.1589 ±0.0004 | 12128 ±39 | 11817 ±41 | 509.6 ±2.6 | |
| | B | PT30 | 1035 ±1 | 25456 ±510 | 23 ±0 | 491.6 ±1.7 | 0.1823 ±0.0006 | 14051 ±50 | 12093 ±102 | 508.8 ±1.7 | |
| | B | PT31 | 1092 ±1 | 30804 ±617 | 20 ±0 | 492.7 ±1.9 | 0.1853 ±0.0004 | 14281 ±39 | 12040 ±109 | 509.8 ±2.0 | |
| | B | PT32 | 1072 ±2 | 52299 ±1049 | 13 ±0 | 490.9 ±2.4 | 0.2117 ±0.0008 | 16493 ±75 | 12646 ±188 | 508.9 ±2.5 | 3.4 ±0.1 |
| | C | PT35 | 1198 ±2 | 36637 ±730 | 19 ±0 | 488.0 ±3.0 | 0.1917 ±0.0007 | 14857 ±69 | 12444 ±128 | 505.6 ±3.1 | |
| | C | PT33 | 1112 ±1 | 102229 ±2047 | 8 ±0 | 478.7 ±2.0 | 0.2320 ±0.0004 | 18372 ±47 | 11179 ±324 | 494.2 ±2.1 | |
| | C | PT34 | 1087 ±1 | 54301 ±1088 | 13 ±0 | 487.8 ±2.0 | 0.2100 ±0.0004 | 16387 ±44 | 12441 ±183 | 505.3 ±2.1 | |
| 2 | A | PT12 | 3843 ±5 | 2368 ±48 | 2176 ±44 | 497.5 ±2.1 | 0.4386 ±0.0007 | 36971 ±94 | 36868 ±94 | 552.1 ±2.3 | |
| | A | PT13 | 2324 ±3 | 1843 ±37 | 1692 ±34 | 498.5 ±1.8 | 0.4390 ±0.0008 | 36983 ±96 | 36850 ±97 | 553.3 ±2.0 | |
| | C | PT09 | 3010 ±4 | 6218 ±125 | 316 ±6 | 1038.9 ±2.2 | 0.2137 ±0.0004 | 11899 ±29 | 11640 ±46 | 1073.9 ±2.3 | |
| | C | PT10 | 3340 ±5 | 19264 ±387 | 117 ±2 | 1034.4 ±1.9 | 0.2205 ±0.0006 | 12327 ±36 | 11604 ±105 | 1069.0 ±2.0 | |
| | C | PT11 | 3147 ±4 | 10802 ±217 | 193 ±4 | 1038.0 ±2.0 | 0.2169 ±0.0004 | 12092 ±30 | 11662 ±65 | 1073.0 ±2.1 | |
| | B | PT17 | 1758 ±2 | 1884 ±38 | 596 ±12 | 1042.3 ±1.6 | 0.2090 ±0.0003 | 11602 ±19 | 11467 ±26 | 1076.8 ±1.7 | 7.3 ±0.7 |
| | B | PT16 | 1703 ±2 | 1074 ±22 | 1012 ±20 | 1041.5 ±2.3 | 0.2089 ±0.0003 | 11599 ±24 | 11520 ±26 | 1076.2 ±2.3 | |
| | B | PT15 | 1707 ±2 | 981 ±20 | 1111 ±22 | 1039.3 ±2.7 | 0.2090 ±0.0004 | 11620 ±30 | 11548 ±31 | 1073.9 ±2.8 | |
| 3 | A | PT18 | 2230 ±2 | 1588 ±32 | 1626 ±33 | 151.4 ±1.4 | 0.3790 ±0.0006 | 43058 ±103 | 42967 ±108 | 170.9 ±1.6 | |
| | A | PT19 | 2271 ±2 | 1289 ±26 | 2042 ±41 | 150.2 ±1.3 | 0.3792 ±0.0005 | 43145 ±94 | 43072 ±98 | 169.6 ±1.4 | |
| | A | PT20 | 2537 ±2 | 2889 ±58 | 1020 ±20 | 150.7 ±1.4 | 0.3800 ±0.0005 | 43223 ±98 | 43077 ±111 | 170.2 ±1.6 | |
| | B | PT21 | 1772 ±1 | 1463 ±29 | 1404 ±28 | 150.0 ±1.3 | 0.3794 ±0.0005 | 43168 ±89 | 43062 ±97 | 169.5 ±1.4 | 4.2 ±1.1 |
| | B | PT22 | 1688 ±1 | 2431 ±49 | 807 ±16 | 150.3 ±1.3 | 0.3801 ±0.0005 | 43260 ±95 | 43076 ±117 | 169.8 ±1.5 | |
| | B | PT23 | 1889 ±3 | 2025 ±41 | 1084 ±22 | 150.5 ±1.7 | 0.3801 ±0.0009 | 43254 ±151 | 43117 ±159 | 170.0 ±1.9 | |
| | C | PT24 | 1112 ±2 | 10208 ±205 | 130 ±3 | 150.5 ±1.6 | 0.3893 ±0.0010 | 44530 ±162 | 43029 ±178 | 170.0 ±1.9 | |
| | C | PT25 | 1484 ±3 | 92 ±5 | 18343 ±974 | 150.1 ±1.6 | 0.3736 ±0.0009 | 42366 ±147 | 42356 ±147 | 169.2 ±1.8 | 5.4 ±0.2 |
| | C | PT26 | 1547 ±3 | 168 ±5 | 10505 ±313 | 150.1 ±1.7 | 0.3729 ±0.0009 | 42269 ±144 | 42251 ±144 | 169.1 ±1.9 | |

$^{\#}\delta^{234}U = ((^{234}U/^{238}U) - 1)*1000.$




### 4.2.2 Patch 2

Patch 2 also contains morphologies of variable relative surface area (suppl. fig. 2). Samples of morphology A are translucent with a brownish hue and tend to form smooth, euhedral crystal faces. Morphology B is milky-white and shows a high degree of non-crystallographic branching close to spherulitic growth. Morphology C also has a milky-white colour but is characterised

by complex aggregates of blocky crystals with a large relative surface area per grain.

As for patch 1, the uncorrected ages vary for each morphological type (table 1). Types B and C cluster around 11,600 and 12,100 BCE respectively, while type A yielded much older ages around 37,000 BCE. Compared to patch 1, samples generally have higher $(^{230}Th/^{232}Th)_t$ values (117 ±2 to 5,994 ±121) indicating relatively low $^{230}Th_0$. Thus, correction for $^{230}Th_0$ yields a much narrower PDF sum (fig. 3) and sub-sample ages for type A and B are identical within the uncertainty of each respective

morphology. Again, a relationship between morphology and $(^{230}Th/^{232}Th)_t$ can be observed, where samples with a smaller relative surface area (morphologies A and B) are less affected by $^{230}Th_0$ contamination than the most complex morphology (type C).

The dating results show that there are two distinct generations of CCC present in this patch (table 1, fig. 4). Types B and C plot along a common regression line (r2.1, $R^2 = 0.98$), whereas type A samples are offset. For types B and C, the resulting

isochron yields an age of 11,524 ±85 BCE, indistinguishable from the PDF peak (fig. 3), and a $(^{230}Th/^{232}Th)_0$ activity ratio of 7.3 ±0.7.

Note that the isochron-derived value for $(^{230}Th/^{232}Th)_0$ is approximately 9 times higher than bulk-earth, meaning that the correction for $^{230}Th_0$ is proportionally larger than for patch 1, especially for samples with even moderately elevated $(^{230}Th/^{232}Th)_t$. Previous U-series dating studies have proposed threshold values of $(^{230}Th/^{232}Th)_t > 100$ or even $> 300$ (Li et al.,

1989; Fensterer et al., 2010), above which detrital thorium contamination is deemed insignificant to the accuracy of dating results. Results from patch 2 indicate that in cases where the true value of $(^{230}Th/^{232}Th)_0$ is multiple times higher than the bulk-earth estimate, threshold values may need to be adjusted to even higher levels (potentially as high as $(^{230}Th/^{232}Th)_t > 500$ for relatively young samples) in order to achieve the same level of accuracy. We argue that such $(^{230}Th/^{232}Th)_t$ limits should be considered as guides, and that each dataset should be evaluated on a case-by-case basis as acceptable limits will be affected

by age, the relationship between analytical uncertainty and uncertainty in $^{230}Th_0$, and the age accuracy needed to address the question at hand. As presented here, isochrons can be a useful and reliable approach if high-quality, reliably dateable samples with high $(^{230}Th/^{232}Th)_t$ are not available.

### 4.2.3 Patch 3

In patch 3, morphologies of type A and B consist of translucent calcite with a brownish hue (suppl. fig. 3, similar to type A of

patch 2). Type A resembles rhombohedral single crystals with subtle non-crystallographic branching while Type B branches more strongly giving the grains a dumbbell-like shape. Type C is significantly larger (up to 3 cm in diameter) and shows a





transition from the brownish rhombohedral type to milky white calcite with a radial-fibrous texture indicative of high degrees of non-crystallographic branching.

The U and Th concentrations of type A and B are similar within an order of magnitude (table 1) with high $(^{230}Th/^{232}Th)_t$ values
$(807 \pm 16$ to $2,042 \pm 41)$ indicating very clean samples with minimal $^{230}Th_0$ contamination. The resulting uncorrected ages of these morphological types are identical within analytical uncertainty (fig. 3).

In contrast, morphological type C shows an extreme range of $(^{230}Th/^{232}Th)_t$ values. The two large specimens, where aliquots from the core of the samples were analysed (suppl. fig. 3), have extremely high $(^{230}Th/^{232}Th)_t$ (10,505 and 18,343 respectively), yielding a very narrow PDF sum that is indistinguishable from uncorrected ages. The third aliquot however, which was a piece
of surface material chipped off from a larger type C grain, yielded the lowest $(^{230}Th/^{232}Th)_t$ $(130 \pm 3)$ of this patch and hence an uncorrected age ~2000 years older.

For this patch, the PDF sum clearly shows a bimodal distribution of corrected ages (fig. 3). An isochron regression can be fitted to the data when exempting morphology C (r3.1, R² = 0.89). The resulting isochron age of 43,065 $\pm$139 BCE $((^{230}Th/^{232}Th)_0 = 4.2 \pm 1.1$, fig. 4c) agrees with the older PDF peak corresponding to types A and B. The isochron age of type
C (r3.2, 42,297 $\pm$ 109 BCE) also confirms the results of the drilled aliquots and the corresponding $(^{230}Th/^{232}Th)_0$ $(5.4 \pm 0.2)$ is similar to the value obtained from types A and B. However, it is noted that the regression is largely controlled by a single, highly inaccurate sample (PT24). Disregarding this sample, a very steep regression with $(^{230}Th/^{232}Th)_0 = 16.8 \pm 3.5$ (i.e., approx. 21x bulk-earth) would be necessary to connect the two clusters of data. Consequently, it is reasonable to propose two distinct generations of CCC at 43,065 $\pm$139 BCE (type A and B) and 42,297 $\pm$ 109 BCE (type C). The isochron approach is thus able
to also resolve differing formation ages within a single patch.

## 4.3 Implications

In theory, $(^{230}Th/^{232}Th)_0$ values derived from CCC isochrons represent the Th activity of the residual water from which the CCCs precipitated. All of the $(^{230}Th/^{232}Th)_0$ values we found $(2.7 \pm 0.2$ to $7.2 \pm 0.7$, suppl. table 1) are comparable to literature data on common speleothems (e.g., Beck et al., 2001; Hubert et al., 2006; Hoffmann et al., 2010; Carolin et al., 2013; Moseley
et al., 2013; Arienzo et al., 2015; Carolin et al., 2016). However, there is significant variability in $(^{230}Th/^{232}Th)_0$ across the three patches. Other studies have reported a much lower in-cave variability with respect to $(^{230}Th/^{232}Th)$ in sediments, speleothems and drip water (Olley et al., 1997; Kaufman et al., 1998). A possible explanation for this high variability can be entertained by considering the dynamic nature of CCC-forming pools. Munroe et al. (2021) reported that the residual pools holding CCCs in Winter Wonderland Cave show different degrees of discolouration of the residual water, indicating that
progressive freezing altered the physio-chemical properties of the residual water. One possible explanation could be that as colloids build-up within the residual water, the ratio between the two main sources of $^{230}Th_0$ (i.e., (i) colloids and (ii) silicates and iron (oxyhydr)oxides; (Short et al., 1988; Henderson and Slowey, 2000; Dorale et al., 2004) changes, thereby altering the Th isotopic composition. A non-linear behaviour of $(^{230}Th/^{232}Th)_0$ could also explain why the isochron approach in patch 1





agrees better with the PDF sum, when sample PT33 is excluded from the regression. At this point however, it remains unclear whether progressive freezing alters $(^{230}\text{Th}/^{232}\text{Th})_0$.

We have shown that for distinct patches of CCCs, an apparent spread in corrected ages can be an artefact arising from underestimating $(^{230}\text{Th}/^{232}\text{Th})_0$ in samples with substantial $^{230}\text{Th}_0$. The data indicate that the relative surface area of a CCC grain correlates with $^{232}\text{Th}$ content, likely because more detrital material adsorbs onto a larger surface. Thus, CCCs with smaller surface areas will tend to require smaller corrections for $^{230}\text{Th}_0$ and yield more accurate ages. Further evidence supporting this is provided by the drilled aliquots of core material from patch 3, which show the highest $(^{230}\text{Th}/^{232}\text{Th})_t$ of all analyses and consequently yielded ages that are not sensitive to correction.

The dating issues that we describe in this study are ultimately tied to sample preparation. Grain sizes of CCCs are highly variable and also seem to vary geographically. Very large specimens, sometimes over 10 cm in size, seem to be more prevalent in Russian caves (e.g., Chaykovskiy and Kadebskaya, 2015 and Y. Dublyansky, unpublished data) while studies from central Europe and the European Alps report mostly mm-sized crystals (Luetscher et al., 2013; Spötl and Cheng, 2014; Richter et al., 2014; Richter et al., 2015; Pavuza and Spötl, 2017; Richter et al., 2017; Colucci et al., 2017; Žák et al., 2018; Richter et al., 2019; Koltai et al., 2021; Richter et al., 2021; Spötl et al., 2021; Kluge et al., 2014). Drilling aliquots to obtain core material from small grains is unfeasible in practise, which consequently demands a critical assessment method. If high-quality samples with high $(^{230}\text{Th}/^{232}\text{Th})_t$ are not available, dating multiple morphological types (if present) and constructing isochrons may be a key strategy in obtaining accurate and precise CCC chronologies.

## 5 Conclusions

- The morphological variability of CCCs in Water Icicle Close Cavern arises from aggregation and varying degrees of non-crystallographic branching, which in turn reflect changes in the chemical environment during the CCC-forming freezing process. Samples range from rhombohedral single crystals that form early in the process to highly branched crystal aggregates that form when the driving forces of crystallisation increased at a later stage.

- Patches of CCCs likely formed during a single freezing event if a) the CCC morphologies show progressive non-crystallographic branching, and b) aliquots define a highly correlated isochron.

- An apparent spread in uncorrected ages can arise from varying degrees of contamination by $^{230}\text{Th}_0$. To guarantee high accuracy of samples with substantial $^{230}\text{Th}_0$, we recommend that ages from a single patch are further analysed using an isochron approach from which both the age of formation and $(^{230}\text{Th}/^{232}\text{Th})_0$ can be calculated. Furthermore, age offsets can be minimised by avoiding surface material where grain size permits.

- To reliably determine the age (range) of a patch of CCCs it is necessary to analyse multiple CCC individuals and – if present – multiple morphologies.

These conclusions have implications for the application of CCCs in palaeoclimatology. If confirmed by studies of CCCs from other caves, the accuracy of some previously published $^{230}\text{Th}/\text{U}$ results (corrected with nominal values of $(^{230}\text{Th}/^{232}\text{Th})_0$) should



be viewed with caution. Of particular concern are the ages reported for those studies for which reported values of $(^{230}Th/^{232}Th)_t$ are low. As a consequence, the climatic interpretation of some studies that used CCCs to reconstruct past permafrost conditions may need to be re-assessed. This applies in particular to studies that analysed whole grains as opposed to aliquots of core material. In addition to increasing the number of aliquots per CCC patch, we recommend applying isochron dating to increase

the robustness of CCC-based palaeoclimate interpretations.

## 6 Code/Data availability

All data necessary to reproduce the results presented herein are provided in the text and supplementary material.

## 7 Author Contribution

PT performed analyses and prepared the manuscript. SDS and GEM performed additional calculations and contributed to
manuscript preparation. XL contributed with additional $^{230}Th/U$ analyses. RLE, CS, YD and JG were instrumental in the study design, directing analyses and preparing the manuscript.

## 8 Competing Interests

The authors declare no competing interests

## 9 Acknowledgements

The authors extend their gratitude to Alan Brentnall for cave access and guiding, Natural England for permitting fieldwork and sampling, and Peter Schroedl and Dylan Parmenter for assistance with $^{230}Th/U$ analyses at the University of Minnesota. We also thank Andy Freem for discovering  the CCCs in Water Icicle Close Cavern as well as Robbie Shone for supporting field work, photography and discovering additional CCC samples.

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
