# Peer review of "230Th/U Isochron Dating of Cryogenic Cave Carbonates"

_Geochronology, 2022_

## Referee Comment (RC1)

[revised manuscript text omitted]

---

## Author Response (AR1)

**Author's Response**

**Response to Review Comments 1:**

Firstly, we would like to thank Karel Žák for taking the time to provide us with this very helpful feedback – it is greatly appreciated. Below is our response to the review comments in detail:

- *General Comments:*

  *One of assumptions on which is based this study is formation of all CCC particles in one patch within one water freezing event. The reasoning for this assumption is based on the occurrence of typical sequence of CCC morphological types, each characterized by carbon and oxygen stable isotope data in an explainable logical sequence, which is in accord with known principles of stable isotope fractionation under water freezing conditions. This assumption is most probably valid, nevertheless, certain uncertainty remains. Under oscillating climate of the Last Glacial similar climatic conditions could occur repeatedly and all other factors are constant or highly similar – cavity morphology, its depth below the surface, lithology and chemistry of the limestone, characteristics of the epikarst zone, etc. We can therefore speculate that repeated freezing conditions with similar morphology of the ice fill the cave and similar chemistry of water which freezes could produce repeatedly the same sequence of morphological types with similar evolution of C and O stable isotope data in the carbonate. After ice melting the products of two or more freezing events could have been deposited together in one patch. An approach which can possibly shed some light on this can be detailed mapping of size distribution and distribution of individual CCC morphological types within each of studied patches together with detailed mapping of patches (giving their dimensions and shapes).*

We absolutely agree that similar CCC-forming conditions could have occurred repeatedly at a given site and produced similar CCC morphologies with similar trends in trace element and stable isotope composition. Note that these chemical and physical properties do not change anymore after CCC precipitated and conserve the original composition. The $^{238}U$-$^{234}U$-$^{230}Th$ system we used in this study, however, evolves continually through radioactive decay. It is highly unlikely that CCCs from different precipitation events would – by accident – form a highly correlated isochron. Our results (especially Patches 2 and 3) show that the isochron approach in fact allows us to reliably differentiate multiple formation events. As is stated in line 55, we do not *assume* but *test the hypothesis* that individual CCC particles form within achievable dating precision. To address this comment, we consider rephrasing the statement in line 55 to state more clearly that we do not make an assumption.

- *Since the study site is presented as a CCC locality for the first time in the international literature, the position of the studied sites within the cave should be presented in more detail, either in a cave map (can be included in an electronic supplement) or by coordinates and/or description in the text.*

In accordance with this comment and others that regard the study site, we will add an extended site description to the supplementary material and add a reference to the main text (line 66).

- *I also consider as important to specify in the discussion from which direction the water producing the CCCs most probably entered the cave. Was it dripping water or flowing water from some kind of periodic water stream? This has some consequences regarding the transport of clay particles within the cave and thus for the interpretation of Th contamination of the samples.*

We are afraid there is no way of knowing this with the evidence we have available. In [1], there is some description of a paleo-flow of water towards the north during a phreatic phase as indicated by scallops on the cave walls. However, this phreatic phase unequivocally predates the (vadose) phase when CCCs formed. Besides that, there are no good indicators of where water was flowing. It seems more reasonable to assume that the water was dripping from the ceiling onto pre-existing ice bodies, rather than a stream pooling and freezing, but there is no direct field evidence available. We will point out those aspects in the extended site description we will add them to the supplementary material.

[1] Gunn, J.; Fairchild, I.J.; Moseley, G.E.; Töchterle, P.; Ashley, K.E.; Hellstrom, J.; Edwards, R.L. (2020): Palaeoenvironments in the central White Peak District (Derbyshire, UK): evidence from Water Icicle Close Cavern. *Cave and Karst Science, 47*, 153–168.

- *It would be useful to compare the levels of clastic Th contamination of the studied samples with 230Th/232Th ratios of other studied and published localities.*

Many authors only published the final ages without providing these additional data. Also, a detailed description of the sample preparation (e.g. cleaning steps and if samples were drilled or digested as a whole) are usually lacking, which makes it difficult to compare these published data. From those papers that did provide $^{230}$Th/$^{232}$Th ratios, all analyses fall within the range we observed in this study (i.e. ~10 - 10,000), and often times show large variation of 2-3 orders of magnitudes. We did not mention these studies explicitly, but this might indicate that these studies possibly fell into the trap of reporting multiple CCC formation periods because of correcting contaminated samples with a poorly chosen $^{230}$Th/$^{232}$Th initial ratio. We will cite a selection of reported values for $^{230}$Th/$^{232}$Th in the discussion part to address this point.

- *Detailed comments*

   *The studied site - Please give an information here about the cave chimneys directed toward the surface. Are they all sediment or limestone boulder blocked? The cave entrance was artificially excavated? What have possibly been the circumstances of cave connection toward the surface in the Last Glacial?*

We will add this information to the extended site description.

- *Results - How do you know it is only the calcite? Based on crystal morphology only? There is no information about XRD mineral identification in the Methods.*

We have XRD data for several specimens, but not for most of the samples of the presented data set. We will remove the respective sentence in line 129 because indeed, we cannot say for sure.

- *References*

   *The list of references is complete and there is no extra reference. The guide for authors is not followed completely. The journal names should be abbreviated and the order of references should follow the guide for authors (copied from this source):*

   *- Single author papers: chronologically, beginning with the oldest. If there is more than one paper in the same year, a letter (a, b, c) is added to the year, both in the in-text citation as well as in the reference list.*

   *- Co-author papers: first alphabetically according to the second author's last name, and then chronologically within each set of co-authors. If there is more than one paper in the same year per set of co-authors, a letter (a, b, c) is added to the year both in the in-text citation as well as in the reference list.*

We will change the reference list to chronological ordering and abbreviate all journal titles – thanks for taking the time to also check the reference list!

**Comments from the main text (gchron-2022-10-RC1-supplement.pdf):**

Line 34: Good point. We'll specify that we're talking about karst water.

Line 36: "speleothem" added

Line 40: An error in the literature database – corrected now.

Line 65: We will add this information to the extended site description of the supplement, but we would like to emphasise that it was our intention to reduce cave-specific aspects and focus on the geochronological methodology.

Line 80: We will add this information to the extended site description.

Figure 2: added VPDB to axis labels

Line 129: see above – will be removed.

Line 199: The abbreviation PDF (probability density function) is introduced in the methods section in line 113.

Line 267: Correct. We will add this information "[...] including all sources of Thorium (such as solutes, colloids and particulate matter)".

Line 289: We'll rephrase "Russian" to "eastern European and Siberian".

Line 301: Actually, both patch 2 and 3 show evidence for at least two freezing events. This is shown by the offset from the regression lines in figure 4. We believe we made this clear in Results and Discussion.

References: see above – will be changed in the revised manuscript

**Response to Review Comments 2:**

We want to cordially thank Dr. David Richards for taking the time to provide these very helpful comments. Our response in detail:

- *[...]*

    *For consideration.*

    *Line 39. "In nature, values for (230Th/232Th)0 vary by orders of magnitude". Please expand on this to report on the sources (and partitioning) of common and radiogenic 230Th, dissolved, detrital and colloidal.*

    We will add a sentence on different sources and phases of Thorium after this line.

- *Line 109: BCE is equivalent to BC.. you can't redefine the zero point or datum CE. Use of BP where present is defined aa 1950 CE is preferred.*

    Makes sense. We'll change all ages to BP.

- *Line 114: Please address the burgeoning literature associated with probability density functions, kernel density estimates (with detrital zircons, speleothem ages, 10Be moraines etc.). Can you justify your illustration of the statistical distribution? I appreciate that your pdfs are the result of a sensitivity analysis, but they may over-emphasize the relative frequencies.*

    This is a very good point but addressing it needs a bit of discussion:

    Our intention with the PDFs was mainly to **visualise the possible range of corrected ages**, based on "reasonable" $(^{230}\mathrm{Th}/^{232}\mathrm{Th})_0$ correction factors for the presented cave setting. We appreciate that KDEs are now preferred in many fields of geochronology (OSL, fission track, etc.), so it should be explained why we do not use them here:

    From our understanding of the literature, the bandwidth of KDEs is determined as a function of the overall distribution of central values (e.g. using the Silverman [1] or Scott [2] method), but not the analytical uncertainty of individual measurements. To incorporate uncertainty, there is the possibility to apply monte-carlo based subsampling and choose a user-defined bandwidth [3], but this approach is more suitable for larger data sets, computationally expensive and not well established yet. For our intents, we thought it would be more appropriate to use PDFs since they allow us to directly control the width of the distribution, which mainly depends on the uncertainty of the $(^{230}\mathrm{Th}/^{232}\mathrm{Th})_0$ correction factor, keeping in mind that Gaussian distributions are an assumption for U/Th ages to begin with.

    For patches 1 and 2, both methods (KDE and PDF) essentially yield the same result, as shown in the figures below:

    Patch 1:

[Figure]

upper panel - uncorrected age in years BP vs measured $^{230}\mathrm{Th}/^{232}\mathrm{Th}$ activity ratio; lower panel - KDE and PDF of corrected ages

Patch 2:

[Figure]

However, for patch 3, the KDE produces an unrealistic bimodal distribution for the very 'clean' age cluster at ~42,300 BP. Because of the very low initial $^{230}$Th, central values of the corrected age ensemble clusters tightly, but the analytical uncertainty of the measurement is not accounted for. PDFs do not have this issue.

Patch 3:

[Figure]

This is the main reason why we chose PDFs over KDEs. We felt this discussion would be distracting to readers and is not essential to the main focus points of the paper: Constructing isochrons is important when dealing with sub-optimal sample material. We are reluctant to add these considerations at length, because we believe it will be distracting to the readers and – as we show above – is of little relevance for this study. We are considering adding a sentence to the respective Method part, stating that "we chose PDFs over KDEs because the former consider the analytical uncertainty directly".

[1] B.W. Silverman, "Density Estimation for Statistics and Data Analysis", Vol. 26, Monographs on Statistics and Applied Probability, Chapman and Hall, London, 1986.

[2] D.W. Scott, "Multivariate Density Estimation: Theory, Practice, and Visualization", John Wiley & Sons, New York, Chicester, 1992.

[3] Weij, R.; Woodhead, J.; Hellstrom, J.; Sniderman, K. An exploration of the utility of speleothem age distributions for palaeoclimate assessment. Quaternary Geochronology 2020, 60, 101112, doi:10.1016/j.quageo.2020.101112.

- *Line 129: Provide evidence for the composition of the CCC, i.e. calcite.*

This was also criticised by reviewer 1 and we will remove this statement for lack of a comprehensive dataset to support the statement.

- *Line 174: Rephrase this statement.. a concentration cannot be expressed as a ratio.*

Agreed.

- *And on this front, please reflect on the use of other indices for detrital components, Al, Zn, Ti etc.. Perhaps future studies could investigate trace elemental variation more fully.*

Indeed, we did not consider using, for example Al, as a proxy for detrital contaminants. This could be a promising approach and we will add a sentence to the implications part to promote it.

- *Line 176. "grouped and sorted ... on a visual basis". Maybe better to say "visually grouped on the basis of relative surface area and roughness".*

  *Expand. What does roughness mean here? "Relative surface area" – per unit volume? Types A, B and C – list here and refer to Supplementary images.*

We will rephrase this part and drop "roughness" because it is redundant.

- *Fig. 4. Why is the 95%CI not 'symmetrical' w.r.t. the regression line? Or do these shaded areas correspond to the confidence intervals for the global data set for each patch, which is then overlain with regressions for subsets.*

We used the seaborn package for Python 3 for these plots, which is a standard tool. The documentation of the linear regression function states that the depicted confidence interval is generated from multiple regression lines of a bootstrapped dataset – hence the asymmetry.

https://seaborn.pydata.org/generated/seaborn.regplot.html

- *Also, your isochron analysis is based on 3-D methods as described in Ludwig and Titterington (1994). Please guide the reader to the tools you have used - Isoplot/Ex, IsoplotR or alternative? How have you approached error correlations?*

We used our own python script, which used the same regression models as IsoplotR. We will add this information at an appropriate location in the text.

Regarding error correlations, we used the least squares method to calculate the regression lines (see line 115). This algorithm does not require error correlations ($\rho$ parameter in IsoplotR), which are only necessary to calculate maximum likelihood regression.

- *Line 184. This is the first mention of (230Th/232Th)t - this should be defined earlier. Is this at time T= t, i.e. time (age) since precipitation T=0, and also referred by others as measured value (m).*

The $(^{230}Th/^{232}Th)_t$ terminology is introduced in line 38. We will change this introduction line to make it more prominent and thorough.

- *Line 209. The data point with most leverage is not an age, it is a data point in isotope space.*

Agreed. We will rephrase it.

- *Table 1. Some uncertainties = 0 in the (230Th/232Th)t column. Please correct.*

Good catch. We will correct it.

- *BCE – please correct. Use age\* and define the datum in \*footnotes. BCE ≠ years before 1950.*

Will change this.

- *Placement of the isochron age derivation in the final column is ambiguous – is this all data for each patch of subset?*

As described in the main text, these are the values of the respective subset. We will change the placement for patch 2 to better represent this.

- *What does isochron-corrected age mean? I presume this is age after correction using derived initial Th for this patch. Please add more detail in footnote. Or remove this column because it is awkward to define these as single ages when they are not independent of the other analyses for the specific patch.*

Agreed, we will delete this column.

- *Line 266. The initial Th ratios calculated do not 'theoretically' represent the values of the 'residual water' from which the CCC precipitated. The water will have more than one source of Th (dissolved, particulate, adsorbed, complexed) and the partitioning between the liquid and solid phase will be complex and variable. The initial Th ratios are representative of the components adsorbed or coprecipitated.*

Agreed. This part will be rephrased.